**www.cambridge.org/gmh**

# Crisis line use and mental health care access among LGBTQ+ young people in the United States

Ronita Nath [ORCID], Derrick D. Matthews and Jonah P. DeChants

The Trevor Project, USA

LGBTQ+ youth; suicide prevention; mental health; crisis services; help-seeking behaviors; 988 Suicide & Crisis Lifeline

**Corresponding author:**
Ronita Nath;
Email: ronita.nath@thetrevorproject.org

## Abstract

Youth who are lesbian, gay, bisexual, transgender, queer or questioning, and other diverse sexual and gender identities (LGBTQ+) experience disproportionately high rates of suicidal thoughts and behaviors compared to heterosexual and cisgender peers, yet many face barriers to care. Data came from a national online survey of 18,663 LGBTQ+ youth aged 13–24 years in the United States (September–December 2023). Analyses focused on participants who reported wanting mental health care in the past year and assessed access, barriers, service modalities and suicidal ideation/attempts. Half of LGBTQ+ youth who wanted mental health care did not receive it. The most common barrier was fear of talking about mental health concerns (42%). Among those who received care, one-on-one therapy was most common (69% in-person and 53% online). Suicidal ideation was lower among youth in therapy (46% in-person and 40% online) compared to those using crisis lines (75%). After adjusting for demographics, hotline use remained strongly associated with elevated risk: adjusted odds ratio (aOR) = 3.77 (95% confidence interval [CI]: 3.11–4.56) for suicidal ideation; aOR = 3.21 (95% CI: 2.62–3.94) for attempts. Despite strong willingness to seek care, structural and identity-related barriers leave many needs unmet. Expanding culturally competent services is essential to reducing suicide risk.

## Impact statement

Young people who are lesbian, gay, bisexual, transgender, queer or questioning, and those with other diverse sexual and gender identities (LGBTQ+) continue to face major barriers when trying to access mental health care, and many who want help do not receive it. In this large national survey, half of LGBTQ+ youth who wanted mental health support in the past year could not get it, and those who did were most likely to access one-on-one therapy. Suicidal thoughts were far more common among young people who relied on crisis lines, reflecting that these services often reach those in greatest distress.

These findings have significant implications for suicide prevention policy and practice. Community-level safety nets – including 24/7 hotlines, text and chat staffed by LGBTQ+-supportive counselors – are essential public health infrastructure. Recent policy changes in the United States that eliminated the LGBTQ+-specific "Press 3" option on the 988 Suicide & Crisis Lifeline reduce access to LGBTQ+-competent care at a time when help-seeking is growing. To save lives, crisis services must be adequately funded, staffed and trained to provide culturally competent support, and specialized pathways for LGBTQ+ youth must be restored and sustained. Ensuring that when LGBTQ+ young people reach out, they connect with someone equipped to help them is both urgent and achievable.

## Introduction

LGBTQ+ young people continue to face disproportionately high rates of suicidal thoughts and behaviors compared to their heterosexual and cisgender peers (Centers for Disease Control and Prevention, 2022). Importantly, this elevated risk does not stem from being LGBTQ+ itself but from the stigma, discrimination and victimization they experience in society, consistent with the minority stress model (Meyer, 2003; Russell and Fish, 2016). Research indicates that many young people experiencing psychological distress and suicidal thoughts often do not seek help; however, those who do are less likely to attempt suicide (Cigularov et al., 2008). Thus, improving help-seeking behaviors among this group is important and could be life-saving. Willingness to seek help has been recognized as a strong protective factor that can prevent the development or worsening of negative health outcomes among youth in general and LGBTQ+ young people in particular (Clement et al., 2015; McDermott and Roen, 2016; McDermott et al., 2018). This willingness to seek professional help for emotional or mental health is associated with an increased likelihood of accessing mental health care in the future (Mojtabai et al., 2016), which in turn can lead

to better mental health outcomes for young people struggling with mental or behavioral health issues (Bear et al., 2020). While willingness to seek help is recognized as a strong protective factor against suicide risk, many LGBTQ+ youth face barriers to accessing timely, supportive mental health care (Burke et al., 2021; Holt et al., 2023). Understanding these barriers and pathways to care is important for informing suicide prevention efforts.

In July 2022, the United States launched the 988 Suicide & Crisis Lifeline, creating a simple three-digit access point for crisis support nationwide (Substance Abuse and Mental Health Services Administration [SAMHSA], n.d.). To address disparities, a dedicated "Press 3" option was introduced in 2022 in partnership with The Trevor Project to connect LGBTQ+ young people with specialized crisis counselors trained in identity-responsive care (The Trevor Project, 2025). The Trevor Project is a US-based nonprofit organization that provides 24/7 crisis intervention and suicide prevention services for LGBTQ+ youth through phone, text and chat, along with research, education and advocacy to advance LGBTQ+ mental health. Evidence indicates that population-specific hotlines can increase help-seeking by providing culturally sensitive care; for example, nearly half of callers to an LGBTQ+ youth crisis line reported they would not have contacted a non-LGBTQ+ line, underscoring the role of tailored services for minoritized groups (Zabelski et al., 2023).

In June 2025, SAMHSA announced that in 30 days it would terminate the "Press 3" option on the 988 Suicide & Crisis Lifeline, shifting LGBTQ+ callers into the general crisis response system (SAMHSA, 2025). The discontinuation of this dedicated channel is concerning, given the well-documented barriers LGBTQ+ youth face when seeking help. These barriers include mental health stigma, fear of negative repercussions, low mental health literacy (i.e., knowledge and beliefs about mental health that facilitate the prevention, identification and management of mental health challenges), difficulties in expressing emotions and a preference for self-reliance (McDermott et al., 2018). Another important barrier – particularly for minors – is parental nonsupport or active prevention of help-seeking, especially when caregivers are unsupportive of a young person's sexual orientation or gender identity. Such gatekeeping can severely limit access to professional mental health services and contribute to heightened distress (Szkody et al., 2025). Even among young people, demographic factors like race, age and gender influence help-seeking behaviors, with people of color, older youth and women being more inclined to seek help compared to those who are White, younger and male, respectively (Hom et al., 2015). Despite the high suicide risk among LGBTQ+ young people, there is a notable lack of research on their help-seeking behaviors. The limited research in this area suggests that LGBTQ+ youth face additional obstacles beyond those encountered by their peers, such as stigma related to their sexual orientation or gender identity and inadequate care that does not address their specific needs (McDermott et al., 2018; Burke et al., 2021; Holt et al., 2023). Negative perceptions of help-seeking among LGBTQ+ youth – for instance, beliefs that school counselors are unhelpful – have been linked to greater suicide risk (Hatchel et al., 2019). Given that young people use a range of services, from outpatient therapy to chat-based and hotline interventions (Mojtabai and Olfson, 2020; Ackerman and Horowitz, 2022), the removal of specialized crisis lines reduces access to one of the few immediately available, tailored supports for LGBTQ+ youth. In response, some states, including Illinois and California, have announced efforts to preserve or expand LGBTQ+-competent crisis support at the state level (California Health and Human Services Agency [CalHHS], 2025; Illinois

Department of Human Services [DHS], 2025). These developments underscore the intersection of structural barriers and shifting policy contexts in shaping the help-seeking landscape for LGBTQ+ young people.

Against this backdrop, our study aimed to examine mental health care access and utilization among LGBTQ+ young people, focusing on who receives care and the types of services accessed, as well as who does not receive care and the barriers they encounter. Based on prior research, we hypothesized that (1) many LGBTQ+ youth who desired mental health care would not have received it; (2) structural and identity-related factors, such as race/ethnicity, socioeconomic status and region, would be associated with lower access to care; and (3) suicidal ideation and attempts would be more common among youth relying on crisis services than among those engaged in ongoing therapy. By situating these findings within the evolving national landscape of suicide prevention policy, this work provides timely evidence to inform interventions and advocacy efforts that seek to ensure inclusive, accessible and effective crisis support for LGBTQ+ youth.

## Methods

### Procedure

Data were drawn from a national online survey on the mental health of LGBTQ+ young people conducted in the United States between September and December 2023. Potential participants were recruited through targeted advertisements on social media platforms (Instagram, Facebook and TikTok). No recruitment occurred through the sponsoring organization's website or social media platforms. After providing informed consent, participants completed an online survey of up to 134 items. A waiver of parental consent was obtained for minors under the age of 18 years, given the minimal risk of the study. To preserve anonymity, no identifying information was collected. To ensure data integrity and minimize the likelihood of automated or fraudulent responses, the survey was hosted on Qualtrics, which includes standard bot-prevention features such as CAPTCHA verification, IP address checks to prevent duplicate submissions and attention filters. The research team additionally reviewed open-ended responses and completion times to identify inconsistent or nonhuman response patterns. For more detailed information on these cleaning procedures and sample screening protocols, see the publicly available methodology of *The Trevor Project's 2024 U.S. National Survey on the Mental Health of LGBTQ+ Young People* (Nath et al., 2024). The study protocol was reviewed and approved by an independent accredited Institutional Review Board (Solutions IRB) on August 16, 2023.

### Participants

Study recruitment was restricted to English- and Spanish-speaking young people of ages 13–24 years in the United States, who identified with a sexual orientation other than heterosexual or a gender identity other than cisgender, and completed *The Trevor Project's 2024 U.S. National Survey on the Mental Health of LGBTQ+ Young People* (Nath et al., 2024). To ensure adequate representation, sample quotas were established by sex assigned at birth and race/ethnicity. Once quotas were met, further responses from those groups were excluded. To be included in the analytic sample, participants had to (a) complete at least 50% of the survey, (b) spend at least 5 min on the survey, (c) pass programmed attention, validity and bot-detection checks and (d) pass a manual

**Table 1.** Demographic characteristics of LGBTQ+ young people who reported wanting mental health care in the past 12 months (N = 15,453)

| Characteristic | n | % | p-value |
|---|---|---|---|
| **Age group (years)** | | | <.001 |
| 13–17 | 7,166 | 46.4 | |
| 18–24 | 8,287 | 53.6 | |
| **Race/ethnicity** | | | <.001 |
| White | 9,466 | 61.3 | |
| Hispanic/Latinx | 1,916 | 12.4 | |
| Black/African American | 1,174 | 7.6 | |
| Asian American/Pacific Islander | 850 | 5.5 | |
| Native/Indigenous | 185 | 1.2 | |
| Middle Eastern/North African | 108 | 0.7 | |
| Multiracial | 1,766 | 11.4 | |
| **Gender identity** | | | <.001 |
| Cisgender girl/woman | 4,021 | 26.0 | |
| Cisgender boy/man | 1,839 | 11.9 | |
| Transgender boy/man | 2,193 | 14.2 | |
| Transgender girl/woman | 1,067 | 6.9 | |
| Nonbinary | 5,332 | 34.5 | |
| Questioning | 1,006 | 6.5 | |
| **Sexual orientation** | | | <.001 |
| Bisexual | 4,297 | 27.8 | |
| Pansexual | 2,598 | 16.8 | |
| Lesbian | 2,534 | 16.4 | |
| Queer | 2,041 | 13.2 | |
| Gay | 1,809 | 11.7 | |
| Asexual | 1,468 | 9.5 | |
| Unsure | 555 | 3.6 | |
| Heterosexual | 151 | 1.0 | |

*Note*: Percentages may not total 100 due to rounding. Ns vary slightly due to missing data.

review of qualitative responses to ensure honest and appropriate responding.

A total of 18,663 LGBTQ+ young people between the ages of 13 and 24 completed the survey. Of these, 18,361 responded to the item on desire for mental health care in the past year. Among them, 15,453 participants (84%) reported wanting psychological or emotional counseling from a counselor or mental health professional, whether or not they ultimately received it. Because this study focuses on access, barriers and experiences of mental health care, analyses are based on this subsample. Sample demographic characteristics are presented in Table 1.

## Measures

Data were drawn from *The Trevor Project's 2024 U.S. National Survey on the Mental Health of LGBTQ+ Young People* (Nath et al., 2024), a cross-sectional online survey. Suicidal ideation and attempts in the past year were assessed using items adapted from the Centers for Disease Control and Prevention's Youth Risk Behavior Survey (CDC, 2023). Access to mental health care was assessed with the question: "In the past 12 months, have you wanted psychological or emotional counseling from a counselor or mental health professional?" Response options included "No," "Yes, but I didn't get it," and "Yes, and I got it." For those who reported an unmet need, barriers to care were measured with the item: "Did you not see a counselor or mental health professional for any of the following reasons?" Participants could endorse multiple nonexclusive options from a list of 23 possible barriers; the 10 most common responses are published on the study's website. For those who accessed care, participants were asked: "In the past 12 months, how have you received your psychological or emotional counseling?" Options included in-person one-on-one therapy, virtual one-on-one therapy, text/chat-based therapy, in-person group therapy, virtual group therapy, hotline/crisis services and "another form of counseling (please specify)." Multiple responses were allowed.

## Data analysis

Chi-square tests were used to assess associations between categorical variables in accordance with our first and second hypotheses on access to care and demographic differences. One-way analyses of variance were employed to evaluate differences in means across groups. Statistical significance was defined as $p < 0.05$ for all analyses. For our third hypothesis on the relationship between mental health care use and suicide risk, multivariable logistic regression was used to examine the relationship between mental health care use and past-year suicide ideation and attempts, while controlling for age, race/ethnicity, gender identity, sexual orientation, socioeconomic status and region.

## Results

### Access to mental health care

Among LGBTQ+ young people who reported wanting psychological or emotional counseling in the past year (84% of the sample), only half (50%) reported receiving it. White LGBTQ+ young people were the most likely to receive it (54%), followed by LGBTQ+ young people who were Native/Indigenous (53%), Middle Eastern/Northern African (52%), Multiracial (49%), Asian American/Pacific Islander (45%), Hispanic/Latinx (42%) and Black/African American (40%). By gender identity, transgender boys and men were the most likely to receive mental health care (54%), followed by transgender girls and women (52%), cisgender girls and women (51%), nonbinary young people (51%), gender-questioning young people (46%) and cisgender boys and men (41%). Across sexual orientation, those identifying as queer were the most likely to receive care (55%), followed by heterosexual transgender youth (53%), bisexual youth (52%), gay or lesbian youth (49%), those not sure of their sexual orientation (49%), asexual youth (48%) and those identifying as pansexual, who were the least likely (46%). LGBTQ+ young people aged 18–24 years were more likely to receive mental health care (52%) than those aged 13–17 years (47%). Participants who reported meeting their basic needs were more likely to receive mental health care (52%) than those unable to meet their basic needs (44%). Regionally, those living in the Northeast were the most likely to access mental health care (56%), followed by those living in the Midwest (52%), the West (50%), and the South (45%).

### Barriers to mental health care

LGBTQ+ young people reported numerous and varied reasons for not receiving desired mental health care. Among participants who wanted but did not receive care, the most frequently reported barrier was fear of talking to someone else about mental health concerns (42%). Other common barriers included affordability (40%), needing parental or caregiver permission (37%), fear of not being taken seriously (34%) and concern that seeking help would result in someone calling the police or having them involuntarily hospitalized (31%).

Patterns of barriers differed across demographic groups. The second most common reason varied by race/ethnicity: White LGBTQ+ young people more often cited affordability (41% vs. 38%), whereas LGBTQ+ young people of color more often cited needing parental or caregiver permission (41% vs. 34%). Barriers varied most widely by gender identity (Figure 1): LGBTQ+ young people who were transgender, nonbinary or gender-questioning were more likely to report barriers related to worries about not being taken seriously, or fears that accessing care would result in someone calling the police or being involuntarily hospitalized; nearly half (46%) of transgender boys and men expressed this as a reason for not accessing mental health care.

### Types of mental health care

Among LGBTQ+ young people who received mental health care, most reported accessing it via in-person one-on-one therapy (69%) or online one-on-one therapy (53%). Group therapy was less common (in-person 8% and online 3%), as was text or chat-based therapy (6%). Additionally, 10% of LGBTQ+ young people accessed a hotline or crisis service as a form of mental health care, and 3% said they used another form of mental health care that was not listed. Percentages do not add up to 100% as LGBTQ+ young people reported using multiple ways of accessing mental health care. While most LGBTQ+ young people received mental health care via one modality (64%), several used two (25%) or three (8%)

and the remaining used four or more (4%). LGBTQ+ young people who accessed one-on-one therapy, either in-person or online, on average used fewer additional modes of accessing mental health care than those using other forms (Figure 2).

### Suicidal thoughts and behaviors

Suicidal thoughts and behaviors among LGBTQ+ young people who accessed care varied significantly by modality ($p < 0.05$, Figure 3). A minority (24%) of LGBTQ+ young people accessed mental health services outside of in-person or online one-on-one therapy, but these individuals reported higher rates of suicidal thoughts and behaviors in the past year. Notably, 75% of LGBTQ + young people using a hotline or crisis services in the last year reported seriously considering suicide in the past year, compared to 46% of LGBTQ+ young people who attended in-person one-on-one therapy and 40% who attended virtual one-on-one therapy. However, because patterns of service use and suicide risk differ across demographic groups, we conducted a multivariable logistic regression controlling for demographic factors and examining all service modalities simultaneously (Table 2). After these adjustments, the association between suicide ideation and attempts and mental health care modality remained. Notably, those who used a hotline or crisis service in the past year had a 277% higher odds of past-year suicide ideation (adjusted odds ratio [aOR]: 3.77, 95% confidence interval [CI]: 3.11–4.56), and 221% higher odds of a past-year suicide attempt compared to those who did not access these services (aOR: 3.21, 95% CI: 2.62–3.94).

### Discussion

This study provides a comprehensive examination of mental health care access, barriers and outcomes among LGBTQ+ young people in the United States. Consistent with prior work (James et al., 2016),

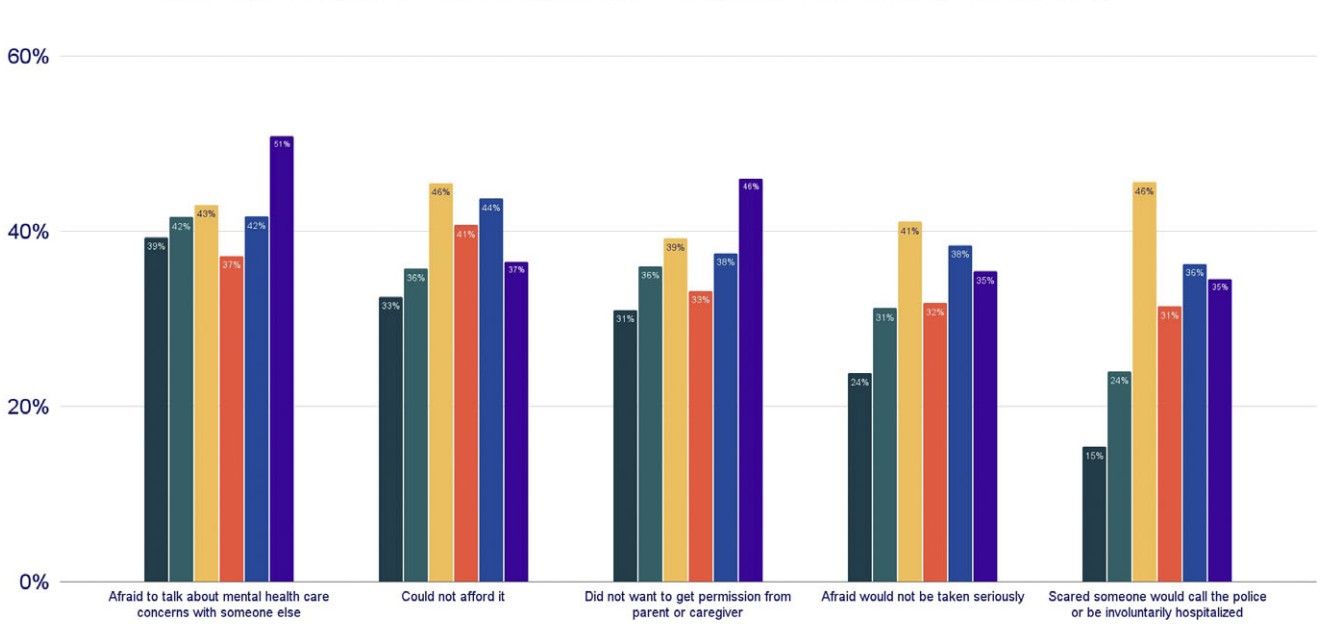

**Figure 1. Percentage of LGBTQ+ young people endorsing the top five barriers to accessing desired mental health care in the past 12 months, by gender identity.** *Percentages reflect the proportion of respondents within each gender identity category who endorsed each barrier. Fear of police involvement or involuntary hospitalization was especially common among transgender boys/men (46%), while affordability was more frequently cited by transgender boys/men (46%) and nonbinary youth (44%). Gender-questioning youth most often reported fear of talking about their mental health care concerns with someone else as a barrier (46%).*
*Note: Participants could select multiple responses; percentages do not total 100.*

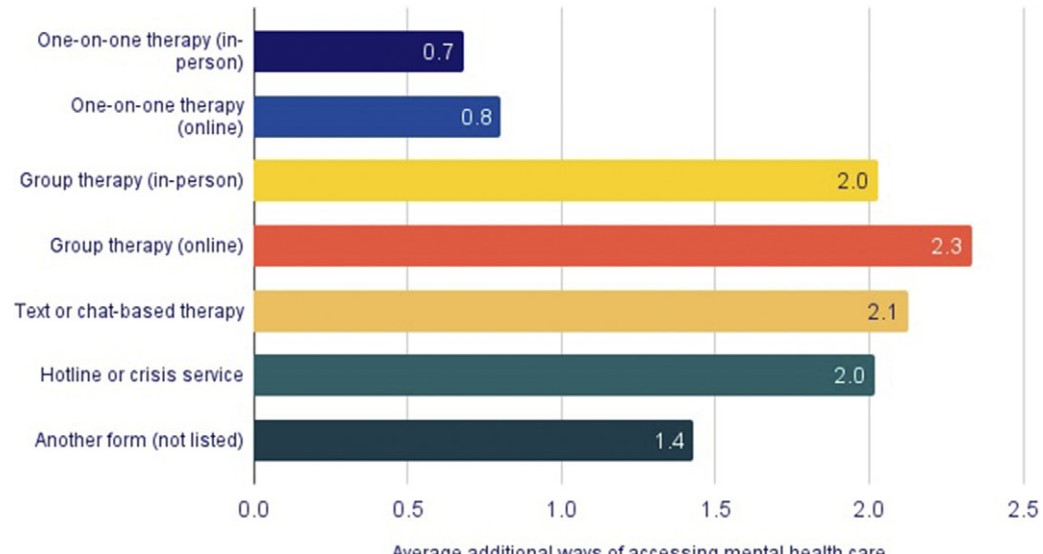

**Figure 2. Average number of additional modalities of mental health care accessed by LGBTQ+ young people, by primary modality of care.** *On average, young people who accessed in-person (0.7) or online (0.8) one-on-one therapy reported fewer additional modalities than those who accessed other types of care. In contrast, youth using online group therapy (2.3), text or chat-based therapy (2.1), or hotline/crisis services (2.0) reported the highest number of additional modalities, reflecting more complex or urgent help-seeking patterns.*
*Note:* "Additional modalities" refers to the number of other forms of care accessed in the past 12 months beyond the primary modality listed.

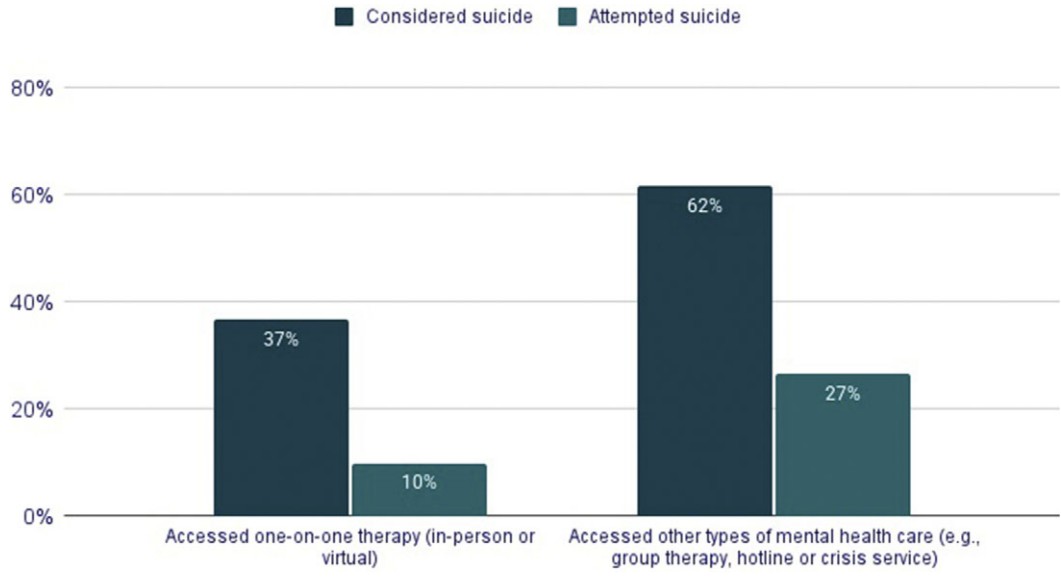

**Figure 3. Past-year suicidal ideation and suicide attempts among LGBTQ+ young people, by type of mental health care accessed.** *Youth who accessed one-on-one therapy (in-person or online) reported lower rates of suicidal ideation (37%) and attempts (10%) compared to those who accessed other types of mental health care (62% ideation; 27% attempts). Differences across modalities were statistically significant at p < 0.05.*

we found that a substantial proportion of LGBTQ+ youth wanted mental health care in the past year but did not receive it. Those who did access care were most likely to do so via one-on-one therapy, either in person or online. Access to one-on-one therapy was uneven across demographic groups, although youth of color and those unable to meet basic needs were significantly less likely to receive services, reflecting broader structural inequities in US mental health care systems (Mongelli et al., 2020). Youth who accessed one-on-one therapy reported lower rates of suicidal ideation and attempts compared to those using crisis lines or other services. LGBTQ+ young people who relied on hotlines or crisis services

demonstrated the highest levels of suicidal ideation and attempts, underscoring that these modalities are often used by those in the most acute distress. Importantly, even after controlling for demographic factors, hotline use remained strongly associated with elevated risk: youth who accessed these services had nearly four times the odds of reporting suicidal ideation and more than three times the odds of a suicide attempt compared to peers who did not.

Taken together, these results highlight several implications for clinical practice and suicide prevention efforts. First, access to sustained, LGBTQ+-competent one-on-one therapy remains critical but must be made more equitably accessible across populations.

**Table 2.** Association between types of mental health care utilization and past-year suicide ideation and attempts (*N* = 7,720)

|  | Adjusted odds ratio (95% Confidence interval) | |
| --- | --- | --- |
|  | Past-year suicide ideation | Past-year suicide attempt |
| One-on-one therapy (in-person) | 1.22 (1.07, 1.39) | 1.25 (1.02, 1.52) |
| One-on-one therapy (online) | 0.90 (0.80, 1.02) | 0.93 (0.78, 1.10) |
| Group therapy (in-person) | 2.11 (1.73, 2.59) | 2.78 (2.23, 3.46) |
| Group therapy (online) | 1.30 (0.97, 1.76) | 1.56 (1.09, 2.23) |
| Text/chat-based therapy | 1.29 (1.02, 1.64) | 1.26 (0.94, 1.68) |
| Hotline or crisis service | 3.77 (3.11, 4.56) | 3.21 (2.62, 3.94) |

*Note*: Model controls for age, race/ethnicity, gender identity, sexual orientation, socioeconomic status and geographic region.

Second, crisis services remain essential but must be understood as part of a continuum. Higher suicide risk among those using hotlines is not evidence of ineffectiveness but rather that these services are often the last line of support for youth at greatest risk (Zabelski et al., 2023). Given the alarming increase in youth suicide rates and the higher rates of suicidal thoughts and behaviors in LGBTQ+ young people compared to their heterosexual cisgender peers (Centers for Disease Control and Prevention, 2022), it is imperative that we continue to develop and implement evidence-based suicide risk identification and intervention strategies that specifically encourage help-seeking among LGBTQ+ people. Although this study focuses on the United States, its implications extend globally. Similar LGBTQ+-affirming mental health and crisis support services in other countries – such as MindOut in the United Kingdom and QLife in Australia – demonstrate that culturally competent, community-based approaches can improve help-seeking and well-being among LGBTQ+ populations (MindOut, 2011; QLife, 2024). Evaluations of these programs indicate that when LGBTQ+ people can connect with trained supporters who understand their identities and experiences, they report greater comfort, trust and emotional relief (MindOut, 2011; QLife, 2024). Conversely, evidence from both evaluations and broader policy contexts suggests that when specialized services are unavailable, underfunded or perceived as unsafe, vulnerable LGBTQ+ individuals may disengage from crisis systems altogether (MindOut, 2011; Zabelski et al., 2023; QLife, 2024). Thus, strengthening LGBTQ+-inclusive crisis response infrastructure is not only a US concern but a global priority for suicide prevention. Effective suicide prevention requires a multifaceted approach, which includes integrating suicide prevention into health care systems, community-based crisis services and utilizing new technologies for real-time risk monitoring (Arango et al., 2021). These strategies are essential to make help-seeking more accessible and effective. The National Strategy for Suicide Prevention emphasizes the importance of a systematic approach to suicide care, incorporating evidence-based practices, such as screening, assessment, safety planning and follow-up services, to provide continuous support and intervention for at-risk individuals (U.S. Department of Health and Human Services, 2024). This comprehensive approach ensures that those in need are identified early and connected with appropriate resources, thereby facilitating help-seeking behaviors and reducing suicide risk.

Beyond clinical practice, these findings also carry broader implications for public health and policy. Twenty-four-hour crisis lines, chat and text services staffed by responders who are supportive and understanding of LGBTQ+ identities must be preserved as core public health infrastructure. The July 2025 discontinuation of the national "Press 3" option within the 988 Suicide & Crisis Lifeline, routing LGBTQ+ callers to specialized counselors, removed a dedicated pathway at a time of rising need. While the stated rationale emphasized standardization, the change risks eroding culturally competent, LGBTQ+-responsive care that research suggests is critical for help-seeking among LGBTQ+ youth (Burke et al., 2021; Zabelski et al., 2023). Evidence from The Trevor Project's crisis lines illustrates the scale of this need: in the days following the 2024 election, the organization recorded a 700% spike in crisis contacts (Alfonseca, 2024). This surge signals that young people's desire for help is growing just as many LGBTQ+ programs face cuts or bans, underscoring the urgency of protecting and expanding accessible, confidential services – particularly for younger teens, who remain least likely to secure professional care. States such as Illinois and California have begun developing their own LGBTQ+ crisis response strategies (CalHHS, 2025; Illinois DHS, 2025), but uneven implementation may exacerbate regional disparities. To ensure equity, both federal and state systems should restore and sustain specialized services for at-risk populations within 988, while also guaranteeing that all crisis counselors receive robust training in LGBTQ+ cultural competency. Our findings also highlight the urgent need to reduce barriers to care. The stigma surrounding mental illness and help-seeking behaviors remains a significant barrier to accessing care among LGBTQ+ youth. Being afraid of discussing mental health concerns with someone else was the most frequently cited barrier by LGBTQ+ youth in accessing mental health care. Reducing this stigma is important for improving mental health outcomes and increasing the likelihood that individuals at risk will seek the help they need. Public awareness campaigns, particularly those targeting schools and community settings, can play a vital role in changing perceptions and encouraging help-seeking behaviors. Gatekeeper training programs – teaching teachers, coaches, parents and peers to recognize warning signs, ask directly about suicide and connect youth to care – and psychoeducational content have shown promise in improving knowledge and attitudes toward mental health and suicide, leading to a sustained impact on reducing suicides in communities (Arango et al., 2021). These campaigns should also highlight the role of adults, as not wanting to get permission from a parent or caregiver was a common reason for not accessing care, especially for LGBTQ+ young people of color. These adults may be unaware they are functioning as a barrier to care, but they could become powerful facilitators of not just mental health care access, but mental health in general, if equipped with information about how to discuss mental health with their children. It should be noted, however, that not all fears about accessing services may be about the stigmatization of mental health care. Nearly half (46%) of transgender boys and men who did not receive care cited fear about someone calling the police or being involuntarily hospitalized. This heightened concern among transgender boys and men may be indicative of the societal stigmatization of transgender identities both outside and within mental health care settings. Even still, transgender young people were the most likely to access care in the last year, which may be partially attributable to the medicalization of gender dysphoria and a resulting increased likelihood of interfacing with health care more broadly (Dewey and Gesbeck, 2017). These findings underscore that it is not enough to expand mental health care access; care must also be LGBTQ+-competent, inclusive and responsive to LGBTQ+ identities. Encouraging help-seeking among LGBTQ+

young people is only an effective strategy if sufficient services exist to prevent, diagnose and treat mental health concerns. The expansion of telehealth services has further increased access to mental health care, particularly for populations disproportionately affected by suicide, including LGBTQ+ young people (Waad, 2019). Crisis intervention systems, including mobile crisis units and crisis stabilization centers, ensure timely access to mental health services during a crisis. Making services like the 988 Suicide & Crisis Lifeline easily accessible to everyone is vital, especially when it can be leveraged to provide LGBTQ+-competent care through integrated partnerships with specialized providers, such as TransLifeline and The Trevor Project's crisis services. These services offer culturally competent support tailored to the unique needs of LGBTQ+ young people, reducing barriers to help-seeking and ensuring they receive the care they need in a LGBTQ+-supportive environment. Providing LGBTQ+-competent, culturally responsive services also means addressing the intersectionality of LGBTQ+ identities with other identities, such as race, ethnicity and disability. Integrating culturally competent and specialized care into routine mental health services, including preventive care, ongoing therapy and community-based support, can help address the unique challenges faced by LGBTQ+ young people. By fostering an environment of understanding and acceptance within all areas of mental health care, we can better support LGBTQ+ youth and promote their overall well-being.

## Limitations

This study has some limitations that should be considered. First, it is cross-sectional and cannot establish causal relationships between the type of care accessed and suicide risk. Second, all measures were self-reported and may be subject to recall or social desirability bias. Third, although the survey included items on suicidal ideation and attempts taken from the CDC's Youth Risk Behavior Survey, it did not employ standardized clinical measures such as the Columbia–Suicide Severity Rating Scale for suicide risk or the Suicide Resilience Scale for Adults-18 for protective factors. These instruments offer a more granular assessment of risk and protection and could strengthen predictive validity in future research. Fourth, while the survey employed quotas to enhance representativeness, it was not a probability sample, and findings may not generalize to all LGBTQ+ young people in the United States. Finally, although we examined differences across demographics, we did not explore how intersecting identities (e.g., race and gender identity) uniquely shape access and outcomes. Future work should apply intersectional and longitudinal approaches to better understand trajectories of help-seeking and the long-term impact of different care modalities. Qualitative studies may also provide valuable insight into the lived experiences underlying the quantitative patterns observed here.

## Conclusions

In summary, this study underscores both the urgency and the opportunity to improve mental health care for LGBTQ+ young people. While many youth desire care, structural barriers and inequities limit access, and those at highest risk are often funneled toward crisis services rather than sustained therapy. As national suicide prevention infrastructure evolves, including recent changes to 988, policymakers, providers and advocates must ensure that LGBTQ+ youth can access culturally competent, continuous care that meets their unique needs. Doing so is not only an equity imperative but also a necessary step in reducing suicide risk among one of the nation's most vulnerable populations.

**Open peer review.** To view the open peer review materials for this article, please visit http://doi.org/10.1017/gmh.2025.10099.

**Data availability statement.** In alignment with our ethical obligations to protect the safety, confidentiality and trust of LGBTQ+ youth participants, the data for this study are not publicly available. Further details about study design and measures may be obtained from the corresponding author upon reasonable request.

**Acknowledgments.** The authors express their deep gratitude to the LGBTQ+ young people who participated in this study and shared their experiences. Their willingness to contribute made this research possible.

**Author contribution.** RN and DDM conceived the study, with RN leading the overall project direction. DDM conducted the data analysis. RN and DDM drafted the initial manuscript. JPD contributed to writing, interpretation and critical revision of the manuscript. All authors reviewed, edited and approved the final version of the manuscript and agree to be accountable for all aspects of the work.

**Financial support.** This work was supported by The Trevor Project.

**Competing interests.** The authors declare none.

**Ethics statement.** This study received ethical approval from Solutions IRB (protocol #0136). All research procedures adhered to established standards for research with human participants. Written informed consent was obtained from all participants. For participants under the age of 18 years, Solutions IRB granted a waiver of parental consent due to the study's minimal risk and the recognition that requiring parental permission could place LGBTQ+ youth at risk if they had not disclosed their identity to their parents. No personally identifying information was collected at any stage. The study complied with all applicable guidelines for the ethical conduct of research.

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
