## [Reviewer Report]

Review GMH-2025-0329

Crisis Line Use and Mental Health Care Access Among LGBTQ+ Young People in the United States

I sincerely thank the journal and the authors of this manuscript for allowing me to contribute suggestions to this interesting research. However, there are some aspects that could be improved. In particular:

Abstract, title and Introduction:

1. Please include in the Abstract the changes made to line 988 for non-US readers, including the mean and standard deviation of age, as well as the specific variables that were measured.

2. Do not repeat the same information in the Impact Statement as in the abstract. Only include short sentences at the beginning to introduce the most notable results and leave the text as it is from ‘These findings have important implications...’ onwards.

3. Please include up-to-date references (maximum 5 years old) in the introduction and only include older references on classic authors.

4. Include up-to-date references on seeking help as a protective factor against suicidal behaviour in LGTBIQ+ groups. We recommend consulting the DOI: https://doi.org/10.1007/s12144-024-06611-3

https://doi.org/10.3390/bs14050422

https://doi.org/10.1111/eip.13405

5. Clarify the objective of the study and the possible hypotheses that were initially proposed.

Methods:

6. In the methods section, please divide it into subsections of scientific articles (participants, measures, procedure, data analysis).

7. In participants, include only the description of the study sample (gender, age, and other demographic variables; include a table if necessary). The time of data collection should be included in the procedure, as well as favourable resolution from the ethics committee.

8. Include inclusion and exclusion criteria for the study sample in participants.

9. If only the survey has been applied, include it in measures, and name it in inclusion and exclusion criteria.

10. Organise the method as indicated in point 5.

Results:

11. The demographic data included in the results alongside Table 1 should be included in the participants section. The results may include frequency analysis as descriptive analysis and comparisons between cisgender and LGTBIQ+ samples.

12. Results based solely on proportions and associations should be improved. Include other analyses that may justify a higher level of complexity in the results.

13. Include a sub-section on limitations in the discussion, stating that the results should be taken with caution as they are based exclusively on a survey method. It would be advisable to include brief risk (C-SSRS) and protection (SRSA-18 adults) measures for suicidal behaviour in these guidelines, as they have a higher level of prediction of repeat attempts and studies can be carried out with greater scientific rigour.

Discussion:

14. Just to point out that a sub-section is missing within the discussion of the clinical applications of the results offered by this study. Although they are already mentioned in some paragraphs, it would be advisable to separate them from the discussion of results, and a section on limitations should also be included. Otherwise, this part is excellent.

15. Check the APA 7th ed. regulations in the reference list.

---

## [Reviewer Report]

Thank you for the opportunity to review “Crisis Line Use and Mental Health Care Access Among LGBTQ+ Young People in the United States”. Now more than ever, this is an important topic that warrants further attention. I have a few comments, but overall found this paper to be comprehensive and detailed.

• Introduction: The introduction provides a comprehensive overview of mental healthcare barriers experienced by LGBTQ youth. One barrier I didn’t see mentioned is parents that prevent help-seeking especially if they are unsupportive of the individual’s gender identity/sexual orientation (e.g., Szkody, E., Sotomayor, I., Hobaica, S., Jans, L., Lopez, C., Pinder, J., & Schleider, J. L. (2025). Barriers to Mental Health Support and Recommendations for Improvement From the Perspectives of LGBTQ+ Youth. Journal of Adolescent Health, 77(4), 667–678. https://doi.org/10.1016/j.jadohealth.2025.06.014)

• Methods: With the increasing issues of bots filling out online surveys, I would be interested in a more detailed description of the bot detection check methods and the manual review of qualitative responses. What did each of these cleaning methods entail and how many responses were screened out at this stage?

• Results: for other types of mental healthcare access (open response option), were there consistent responses for this question? It would be interesting to see what other support LGBTQ youth are receiving that they would classify as mental health care to see if other avenues exist outside of one-on-one therapy.

---

## [Editor Report]

Dear Authors, thank you for submitting this manuscript to the Special Issue on self-harm and suicide.

In addition to the Reviewers' comments, which I hope you’re willing to address, I have a few minor comments:

1. With a global readership, not all readers will be familiar with the U.S. context, so please read with the “eye of an outsider' and edit for clarity. For example, the first line of the Background references the 988 Hotline, which is likely to be unfamiliar to many readers (so you could remove that or explain it). Likewise, reference to the “South" (abstract/results) may lack meaning for global readers.

2. Please briefly describe the Trevor Project in the Intro (again, non-US readers may not have heard of the organization); just a sentence that may provide more information (context, services, etc.).

3. In the Discussion, please include some comparison with findings of other similar suicide hotlines in other countries; this will help the reader understand the global relevance. In other words, what can be understood and learned from your study in the global context?

Many thanks, Jerome

---

## [Editor Report]

Dear Authors: Many thanks for addressing my and the Reviewers' comments. This manuscript will make an important contribution to our special issue.